# Sample Adaptive MCMC

**Michael H. Zhu**
Department of Computer Science
Stanford University
Stanford, CA 94305
mhzhu@cs.stanford.edu

## Abstract

For MCMC methods like Metropolis-Hastings, tuning the proposal distribution is important in practice for effective sampling from the target distribution $\pi$. In this paper, we present Sample Adaptive MCMC (SA-MCMC), a MCMC method based on a reversible Markov chain for $\pi^{\otimes N}$ that uses an adaptive proposal distribution based on the current state of $N$ points and a sequential substitution procedure with one new likelihood evaluation per iteration and at most one updated point each iteration. The SA-MCMC proposal distribution automatically adapts within its parametric family to best approximate the target distribution, so in contrast to many existing MCMC methods, SA-MCMC does not require any tuning of the proposal distribution. Instead, SA-MCMC only requires specifying the initial state of $N$ points, which can often be chosen *a priori*, thereby automating the entire sampling procedure with no tuning required. Experimental results demonstrate the fast adaptation and effective sampling of SA-MCMC.

## 1 Introduction

Markov Chain Monte Carlo (MCMC) methods are a large class of sampling-based algorithms that can be applied to solve integration problems in high-dimensional spaces [1]. The goal of MCMC methods is to sample from a probability distribution $\pi(\theta)$ (known up to some normalization constant) by constructing a Markov chain with limiting distribution $\pi(\theta)$ that visits points $\theta$ with a frequency proportional to the corresponding probability $\pi(\theta)$.

For MCMC methods like Metropolis-Hastings [2, 3], the choice of the proposal distribution $q(\cdot|\theta^{(k)})$ is important in practice for effective sampling from the target distribution. Metropolis-Hastings (MH) is generally used with random walk proposals where local moves based on $q(\cdot|\theta^{(k)})$ are used to globally simulate the target distribution $\pi(\theta)$. A suboptimal choice for the scale or shape of the proposal can lead to inefficient sampling, yet the design of an optimal proposal distribution is challenging when the properties of the target distribution are unknown, especially in high-dimensional spaces.

Gelman et al. [4, 5] recommend a two-phase approach where the covariance matrix of the proposal distribution in phase 2 is proportional to the covariance of the posterior samples from phase 1. Adaptive MCMC methods such as Adaptive Metropolis [6] continually adapt the proposal distribution based on the entire history of past states. However, the method is no longer based on a valid Markov chain, so the usual MCMC convergence theorems do not apply, and the validity of the sampler must be proved for each specific algorithm under specific technical assumptions [7, 8]. In this paper, we propose Sample Adaptive MCMC, a method that is adaptive based on the current state of $N$ points and uses an adaptive proposal which is an adaptive approximation of the target distribution.

## 2 Related work

Our substitution procedure is related to the Sample Metropolis-Hastings (SMH) algorithm by Liang et al. [9, ch. 5] and Lewandowski [10]. The SMH algorithm reduces to Metropolis-Hastings for $N = 1$, and our substitution procedure reduces to the method of Barker [11]. The SMH algorithm has also been used by Martino et al. [12, 13] in the context of adaptive importance sampling and with independent SMH proposals by Martino et al. [14] who propose a family of orthogonal parallel MCMC methods where vertical MCMC chains are run in parallel using random-walk proposals and share information using horizontal MCMC steps encompassing all of the chains using independent proposals.

Parallel tempering [15] runs parallel MCMC chains targeting the posterior distribution at different temperatures. Many previous works have studied MCMC methods which simulate from $\pi^{\otimes N}$ (a sample of size $N$ from $\pi$). Early works include the Adaptive Direction Sampler by Gilks et al. [16], the Normal Kernel Coupler by Warnes [17], and the pinball sampler by Mengersen and Robert [18]. In the Normal Kernel Coupler, Warnes [17] first selects one of the $N$ points in the state to update, uses a kernel density estimate constructed from the state of $N$ points to propose a new point, and finally accepts or rejects the proposed swap according to the Metropolis-Hasting acceptance probability.

Goodman and Weare [19] propose an ensemble MCMC sampler with affine invariance. Griffin and Walker [20] present a method for adaptation in MH by letting the joint density be the product of a proposal density and $\pi^{\otimes N}$ and then sampling this augmented density using a Gibbs sampler including a Metropolis step. Their work is related to the works by Cai et al. [21], Keith et al. [22]. Leimkuhler et al. [23] propose an Ensemble Quasi-Newton sampler using gradient information based on time discretization of an SDE that can incorporate covariance information from the other walkers.

The Multiple-Try Metropolis method [24, 25] first proposes $M$ potential candidates, randomly chooses one of the best candidates based on the weights to be the potential move, and finally accepts or rejects the move according to a generalized MH ratio. Neal et al. [26] propose a new Markov chain method for sampling from the posterior of a hidden state sequence in a non-linear dynamical system by first proposing a pool of candidate states and then using DP with an embedded HMM. Tjelmeland [27] describe a general framework for running MCMC with multiple proposals in each iteration and using all proposed states to estimate mean values. Neal [28] propose a MCMC scheme which first stochastically maps the current state $\theta$ to an ensemble $(\theta_1, \ldots, \theta_N)$, applies a MCMC update to the ensemble, and finally stochastically selects a single state. Calderhead [29] presents a general construction for parallelizing MH algorithms.

Population Monte Carlo [30] is an iterated importance sampling scheme with a state of $N$ points where the proposal distribution can be adapted for each point and at each iteration in any way and a resampling step based on the importance sampling weights is used to update the state. Adaptive Importance Sampling [31, 32, 33] represents a class of methods, including PMC, based on importance sampling with adaptive proposals. Our work is also inspired by particle filters [34, 35] and PMCMC [36] which combines standard MCMC methods with a particle filter based inner loop for joint parameter and state estimation in state-space models.

## 3 Sample Adaptive MCMC

We now present the Sample Adaptive MCMC algorithm. Let $p(\theta)$ be the target probability density known up to some normalization constant, and let $\pi(\theta) = p(\theta)/ \int p(\theta')d\theta'$. The state of the SA-MCMC Markov Chain consists of $N$ points at each iteration. We denote the state at iteration $k$ by $S^{(k)} = (\theta_1^{(k)}, \theta_2^{(k)}, \ldots, \theta_N^{(k)})$. Define $\mu(S) = \frac{1}{N} \sum_{n=1}^{N} \theta_n$ to be the mean of the $N$ points in the state $S$. Define $\Sigma(S)$ to be the sample covariance matrix of the $N$ points in the state $S$. Optionally, we can also consider a diagonal approximation of $\Sigma(S)$ that is non-zero along the diagonals and zero elsewhere. When proposing a new point $\theta'$, the proposal distribution $q(\cdot|\mu(S^{(k)}), \Sigma(S^{(k)}))$ is a function of the mean and covariance of all $N$ points in the current state $S^{(k)}$. In our experiments, we use a Gaussian or Gaussian scale-mixture family as our adaptive family of proposal distributions. After proposing $\theta'$, the algorithm might reject the proposed point $\theta'$ or substitute any of the $N$ current points with $\theta'$. For example, the algorithm might substitute $\theta_1^{(k)}$ with $\theta'$ so that the new state becomes $S^{(k+1)} = (\theta', \theta_2^{(k)}, \ldots, \theta_N^{(k)})$. The probabilities of substituting each of the $N$ points with

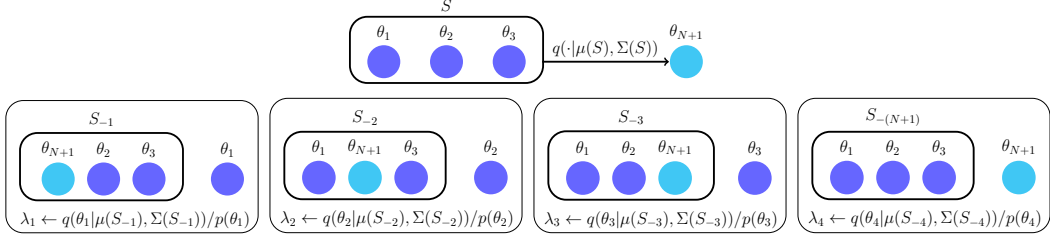

Figure 1: Illustration of one iteration of SA-MCMC for $N = 3$. After the proposed point $\theta_{N+1} \sim q(\cdot|\mu(S), \Sigma(S))$ is sampled, the sets $S_{-1}, \ldots, S_{-(N+1)}$ are used to calculate the substitution probabilities $\lambda_1, \ldots, \lambda_{(N+1)}$. One of the sets $S_{-1}, \ldots, S_{-(N+1)}$ is chosen to be the next state with probability proportional to $\lambda_n$.

---

**Algorithm 1** Sample Adaptive MCMC

---

**Require:** $p(\theta)$, $q_0(\cdot)$, $q(\cdot|\mu(S), \Sigma(S))$, $N$, $\kappa$, $K$
1: Initialize $S^{(0)} \leftarrow (\theta_1, \ldots, \theta_N)$ where $\theta_n \sim q_0(\cdot)$ for $n = 1, \ldots, N$
2: **for** $k = 1$ **to** $\kappa + K$ **do**
3:     Let $S = (\theta_1, \ldots, \theta_N) \leftarrow S^{(k-1)}$
4:     Sample $\theta_{N+1} \sim q(\cdot|\mu(S), \Sigma(S))$
5:     Let $S_{-n} \leftarrow (S$ with $\theta_n$ replaced by $\theta_{N+1})$ for $n = 1, \ldots, N$. Let $S_{-(N+1)} \leftarrow S$.
6:     Let $\lambda_n \leftarrow q(\theta_n|\mu(S_{-n}), \Sigma(S_{-n}))/p(\theta_n)$ for $n = 1, \ldots, N+1$
7:     Sample $j \sim J$ with $\mathbb{P}[J = n] = \lambda_n \big/ \sum_{i=1}^{N+1} \lambda_i, \quad 1 \leq n \leq N+1$
8:     Let $S^{(k)} \leftarrow S_{-j}$
9: **end for**
10: Return $\bigcup_{k=\kappa+1,\ldots,\kappa+K} S^{(k)}$

---

the proposed point and the probability of rejecting the proposed point are all constructed so that the stationary distribution of the SA-MCMC Markov chain is $\pi^{\otimes N}(\theta_1, \ldots, \theta_N) = \prod_{n=1}^{N} \pi(\theta_n)$.

The SA-MCMC algorithm is presented in Algorithm 1 and illustrated in Figure 1. The initialization distribution for initializing the $N$ points is $q_0(\cdot)$. The sets $S_{-n}$, with $\theta_n$ replaced by the proposed point, are the $N + 1$ possibilities for the next state depending on which of the current $N$ points gets replaced, if any. One of the sets $S_{-1}, \ldots, S_{-(N+1)}$ is chosen to be the next state with probability proportional to $\lambda_n$. The number of burn-in iterations is $\kappa$, and the number of estimation iterations is $K$. For any function $h(\theta)$ satisfying $\int |h(\theta)| \pi(\theta)d\theta < \infty$, we can estimate $\int h(\theta)\pi(\theta)d\theta$ by the sample average $\frac{1}{K} \sum_{k=\kappa+1}^{\kappa+K} \frac{1}{N} \sum_{n=1}^{N} h(\theta_n^{(k)})$.

The likelihood ratio $q(\theta_n|\mu(S_{-n}), \Sigma(S_{-n}))/p(\theta_n)$ used to compute the substitution probability $\lambda_n$ corresponds to the inverse of the importance weight used in Metropolis-Hastings, $p(\theta')/q(\theta'|\theta^{(k)})$. Thus, points with low importance weight (i.e. points with low likelihood under the target distribution relative to the proposal) are likely to be replaced by points with higher importance weight. Since the proposed point is compared with the $N$ points in the state before deciding which point to remove, this generally leads to higher acceptance rates compared to having a state with only one point, which is advantageous in problems where evaluating the target density is computationally expensive.

The initialization distribution $q_0$ determines the initial positions of the $N$ points and also the initial mean and scale structure of the proposal distribution. In practice, $q_0$ can be chosen either based on intuition about where the parameters are likely to have high probability under the target distribution or by hyperparameter search. Note that while we must choose $q_0$ carefully depending on the problem, we do not have to tune the proposal distribution. In practice, this usually makes SA-MCMC easier to use since an optimal scale structure for the initialization distribution $q_0$ in SA-MCMC is generally more intuitive than the optimal scale structure (step size) of the random walk proposal in Metropolis-Hastings. In many cases, an optimal choice for $q_0$ can even be chosen *a priori* based on knowledge of the data and the model, thereby automating the entire sampling procedure with no tuning required.

The state of $N$ points, the adaptive proposal distribution $q(\cdot|\mu(S),\Sigma(S))$ based on the current state, and the substitution procedure enable both fast adaptation of the proposal distribution and effective sampling from the target distribution. We first explain how SA-MCMC can transition quickly from its initial state of $N$ points to an empirical representation of the target distribution during the burn-in phase, as illustrated in Figure 2. For example, consider the case where some or even all $N$ points are initialized far away from the high-probability region of the target distribution. In this case, the points $\theta_n$ farthest away from the target distribution will have a much smaller value for $p(\theta_n)$, leading to large $\lambda_n$ and a high probability of substitution. As the points farthest away from the target distribution are replaced, the $N$ points in the state and the corresponding adaptive proposal distribution gradually narrow in on or shift towards the high-probability region of the target distribution. As another example, consider the case where the initial mean is specified correctly but the initial variance is too small. In this case, the points in the center will have a high probability of substitution since the points in the center will have a much larger value for $q(\theta_n)$ compared to points at the end (while values for $p(\theta_n)$ are comparable), so that the variance of the $N$ points in the state gradually increases to the variance of the target distribution. Thus, we see that the form of the substitution probability enables the initial state of $N$ points to adapt to the target distribution under many different initial conditions.

After this burn-in phase, the $N$ points in the state form an empirical representation of the target distribution, and the proposal distribution approximates the target distribution. Using our form for the proposal distribution, when the $N$ points in the state represent a mode of the target distribution, $\mu(S)$ approximates the mean and $\Sigma(S)$ approximates the covariance structure of the mode of the target distribution. As the shape of the proposal distribution approximates the shape of the mode of the target distribution, this enables very effective sampling. Since the substitution probability $\lambda_n$ is an inverse importance weight, $p(\theta_n)$ favors keeping points closer to the mode of the target distribution while $q(\theta_n|\mu(S_{-n}),\Sigma(S_{-n}))$ favors keeping points farther away from the mode of the proposal distribution relative to its covariance $\Sigma(S_{-n})$, balancing each other to ensure that the $N$ points in the state are distributed according to and are approximate samples from the target distribution.

**Theory** Let $\pi(\theta) = p(\theta)/\int p(\theta')d\theta'$ be the target density. Proposition 1 demonstrates that the SA-MCMC Markov chain satisfies the detailed balance condition with respect to $\pi^{\otimes N}(\theta_1,\ldots,\theta_N) = \prod_{n=1}^{N}\pi(\theta_n)$, thus establishing $\pi^{\otimes N}$ as the stationary density of the chain. We then prove that under general conditions on the target distribution and a family of proposal distributions with diagonal covariance matrices, SA-MCMC using a diagonal covariance matrix is ergodic, allowing us to prove convergence in total variation norm to $\pi^{\otimes N}$ and the law of large numbers for estimating expectations with respect to $\pi$ by sample averages. The convergence guarantees for SA-MCMC are proven under the same assumptions on the target distribution as for Metropolis-Hastings. Theorem 1 is based on the theorem of Athreya et al. [37] and the textbook by Robert and Casella [38]. The proofs are given in Appendix 1 and 2. We note that our detailed balance proof is closely related to the detailed balance proof for Sample Metropolis-Hastings given by Lewandowski [10] and Martino et al. [12].

**Proposition 1.** *The SA-MCMC Markov chain from Algorithm 1 with target density $\pi(\theta) = p(\theta)/\int p(\theta')d\theta'$ satisfies the detailed balance condition with respect to $\pi^{\otimes N}(\theta_1,\ldots,\theta_N) = \prod_{n=1}^{N}\pi(\theta_n)$. Hence, $\pi^{\otimes N}$ is the stationary density of the chain, and the chain is reversible.*

**Theorem 1.** *Let $\{S^{(k)}\}$ be the SA-MCMC Markov chain with diagonal covariance matrix from Algorithm 1 with target density $\pi(\theta) = p(\theta)/\int p(\theta')d\theta'$, proposal density $q(\cdot|\mu(s),\mathrm{diag}(\Sigma(s)))$, and $N \geq 3$. Denote the conditional density of $S^{(k)}$ given $S^{(0)}$ by $f_k(\cdot|\cdot)$. Let $h(\theta)$ be any function satisfying $\int |h(\theta)|\,\pi(\theta)d\theta < \infty$. If*

*(A1) $\pi$ is bounded and positive on every compact set of its support $\mathcal{E} \subseteq \mathbb{R}^d$, and*

*(A2) For all $a, b, \delta > 0$, there exist $\epsilon_1, \epsilon_2 > 0$ such that if $a < \sigma_j < b$ and $|x_j - \mu_j| < \delta$ for $j \in 1,\ldots,d$, then $\epsilon_1 < q(x \mid \mu, \mathrm{diag}(\sigma^2)) < \epsilon_2$,*

*then the SA-MCMC Markov chain is ergodic, and*

*(1) $\lim_{K\to\infty} \sup_C \left|\int_C f_K(s|s_0)ds - \int_C \pi^{\otimes N}(s)ds\right| = 0$ for $\left[\pi^{\otimes N}\right]$-almost all $s_0$, and*

*(2) $\mathbb{P}_{s_0}\left[\lim_{K\to\infty} \frac{1}{K}\sum_{k=1}^{K}\frac{1}{N}\sum_{n=1}^{N} h(\theta_n^{(k)}) = \int h(\theta)\pi(\theta)d\theta\right] = 1$ for $\left[\pi^{\otimes N}\right]$-almost all $s_0$.*

*Remark* 1. The convergence result for Metropolis-Hastings can be proved under the assumptions (A1) and $\exists \epsilon, \delta > 0$ such that if $\|x - y\| < \delta$, then $q(y|x) > \epsilon$ [39, 38]. (A2) is a generalization for a family of proposal distributions with different means and scales (e.g. location-scale families).

Our next theorem is stated in a more general form for a proposal distribution $q(\cdot|\gamma(S))$. We prove the uniform ergodicity of SA-MCMC assuming $q(\theta|\gamma)/\pi(\theta)$ is bounded above and below. The proof is based on the proof of Lemma 1 in the working paper by Chan and Lai [40] and is in Appendix 3.

**Theorem 2.** *Let $\pi$ be a positive target density on the parameter space $\Theta$, and let $\{q(\cdot|\gamma) : \gamma \in \Gamma\}$ be a family of positive proposal densities, with $\Gamma$ a convex Euclidean set. Let $\lambda(\theta|\gamma) = q(\theta|\gamma)/\pi(\theta)$ and let the proposal density be $q(\cdot|\gamma(S))$, where $\gamma(S) = N^{-1} \sum_{\theta \in S} \gamma(\theta)$ for some continuous $\gamma : \Theta \to \Gamma$. Let $f_k$ denote the joint densities of $(\theta_1^{(k)}, \ldots, \theta_N^{(k)})$ and $\pi^{\otimes N}$ the product density of $\pi$ on $\Theta^N$. If there exist constants $0 < a < b < \infty$ such that $a \leq \lambda(\theta|\gamma) \leq b$ for all $\theta \in \Theta$ and $\gamma \in \Gamma$, then $\|f_k - \pi^{\otimes N}\|_{\mathrm{TV}} \leq 2(1 - C)^{\lfloor k/N \rfloor}$ for $C = N!(\frac{a}{(N+1)b})^N a^N$.*

Mengersen and Tweedie [41] prove that the Independent MH (IMH) algorithm with independent proposal distribution $q(\cdot)$ is uniformly ergodic if there exists a constant $\alpha > 0$ such that $q(\theta)/\pi(\theta) \geq \alpha$ for all $\theta \in \Theta$, in which case $\|F^k(\theta^{(0)}, \cdot) - \pi\|_{\mathrm{TV}} \leq 2(1 - \alpha)^k$. Holden et al. [42] propose an Adaptive Independent MH (AIMH) algorithm where the proposal distribution $q_k(\cdot|\mathbf{h}_{k-1})$ at iteration $k$ depends on the history $\mathbf{h}_{k-1}$. To preserve the invariance of the sampler, $\mathbf{h}_k$ must be constructed from $\mathbf{h}_{k-1}$ by appending the previous state of the chain if the transition is accepted and the rejected proposed point if the transition is rejected. They prove that the convergence is geometric if there exists a constant $\alpha > 0$ such that $q_k(\theta|\mathbf{h}_{k-1})/\pi(\theta) \geq \alpha$ for all $\theta, \mathbf{h}_{k-1}, k$. Uniform ergodicity is proven by lower bounding the one-step probability of transitioning to the target density each iteration.

The conditions above for IMH and AIMH essentially require that the proposal densities have uniformly heavier tails than the target. We note that a mixture proposal distribution, with the main distribution and a fat-tailed distribution with a small mixing proportion, can be used as a safeguard to guarantee this lower bound [43, 44, 45]. For our algorithm, in practice, we observe that this condition (i.e. $\lambda(\theta|\gamma) \geq a$) is also crucial for SA-MCMC. In the proof and the corresponding bound, this condition corresponds to the term $a^N$ in $C$. Formally, our proof also requires the assumption $\lambda(\theta|\gamma) \leq b$ to lower bound the acceptance probability of $N$ substitutions by the term $(\frac{a}{(N+1)b})^N$ in $C$ corresponding to Line 7 in Algorithm 1. In practice, we find that this condition is not necessary for effective sampling. A proposed point with significantly larger $\lambda$ is unlikely to be accepted in the first place, and if accepted, the point is likely to be replaced quickly, so the worst-case bound of $(N + 1)b$ in the denominator likely understates the practical performance of the algorithm. To support this conclusion, we conduct extensive experiments on t-distributions with different degrees of freedom in Appendix 4 and observe that only the assumption $\lambda(\theta|\gamma) \geq a$ is necessary in practice.

While IMH and AIMH have weaker assumptions for uniform ergodicity in theory, we note that IMH and AIMH fail to work for any of the examples in our paper since they are not adaptive enough for an independent proposal distribution to work in high-dimensional spaces. We elaborate in Appendix 5.

**Implementation**   A fast, numerically stable implementation of SA-MCMC is given in Appendix 6.

## 4   Experimental results

We first illustrate the adaptive nature of SA-MCMC on toy 1D distributions and then present experimental results for the Bayesian linear regression and Bayesian logistic regression models. Our goal is to sample from the posterior distribution $p(\theta|y)$ of the parameters $\theta$ given the data $y$. We assume only that we can compute $p(\theta|y)$ for any $\theta$ up to a normalization constant; we do not assume any other information or structure of the model as in the setup for Metropolis-Hastings. We will compare the performance of Metropolis-Hastings (MH), Adaptive Metropolis (AM), Multiple-Try Metropolis (MTM), and SA-MCMC (SA). As a benchmark, we also compare to the No-U-Turn Sampler (NUTS) [46] using the implementation in RStan 2.19.2 [47], which is a state-of-the-art Hamiltonian Monte Carlo (HMC) method [48] . Note that unlike all of the other MCMC methods in this paper, NUTS uses the gradient of the target density at every step and is based on discretizations of continuous-time stochastic dynamics.

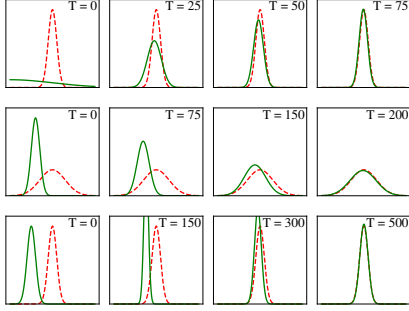

Figure 2: Adaptation of the SA-MCMC proposal distribution (green) to three target distributions (red).

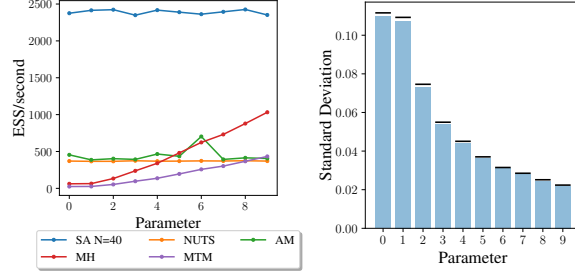

Figure 3: Bayesian linear regression. (***left***) Comparison of ESS/second for each parameter. (***right***) Standard deviation of the SA proposal distribution (blue bar), averaged over iterations, for each parameter compared with the ground truth posterior standard deviation (black line).

We now describe the experimental setup. For Metropolis-Hastings (MH), we use an isotropic normal distribution as the proposal distribution, $q_{\mathrm{MH}}(\cdot|\theta) = \mathcal{N}\left(\theta, \sigma_{q,\mathrm{MH}}^2 \mathbb{I}\right)$, with scale parameter $\sigma_{q,\mathrm{MH}}$, and initialize $\theta^{(0)} \sim q_{0,\mathrm{MH}}(\cdot) = \mathcal{N}\left(0, \sigma_{q,\mathrm{MH}}^2 \mathbb{I}\right)$. We tune $\sigma_{q,\mathrm{MH}}$ to make the acceptance rate close to the optimal value of 23% [49]. For Adaptive Metropolis (AM), we use the optimal MH proposal distribution during the burn-in (non-adaptive) phase and then use the proposal distribution $q_{\mathrm{AM}}(\cdot|\theta^{(1)}, \ldots, \theta^{(k-1)}) = \mathcal{N}\left(\theta^{(k-1)}, s_{\mathrm{AM}}^2 \Sigma^{(k-1)}\right)$ at iteration $k$ with scale parameter $s_{\mathrm{AM}}$ and sample covariance matrix $\Sigma^{(k-1)}$ of the past samples $(\theta^{(1)}, \ldots, \theta^{(k-1)})$. We tune $s_{\mathrm{AM}}$ to make the acceptance rate close to the optimal value of 23%. For Multiple-Try Metropolis (MTM), we use the optimal MH proposal distribution with 3 tries. Finally, for SA-MCMC (SA), we use $q_{0,\mathrm{SA}}(\cdot) = \mathcal{N}\left(0, \sigma_{q_0,\mathrm{SA}}^2 \mathbb{I}\right)$ with scale parameter $\sigma_{q_0,\mathrm{SA}}$ as the distribution for initializing the $N$ starting points. For the proposal distribution, when using the full covariance matrix, we use the Gaussian family $q(\cdot|\mu(S), \Sigma(S)) = \mathcal{N}(\cdot|\mu(S), \Sigma(S))$. When using the diagonal covariance matrix, we use a Gaussian scale-mixture family $q(\cdot|\mu(S), \Sigma(S)) = \sum_i p_i \mathcal{N}(\cdot|\mu(S), c_i \mathrm{diag}(\Sigma(S)))$ with $c = \left[\frac{1}{2}, 1, 2\right]$ and $p = \left[\frac{1}{3}, \frac{1}{3}, \frac{1}{3}\right]$ which we observed works better empirically for logistic regression.

For each of the MCMC methods, we run 16 chains to assess convergence and calculate Effective Sample Size (ESS) divided by the total running time in seconds. For each chain, we run 100,000 burn-in iterations and then collect 1,000,000 samples. For NUTS, we use 10,000 burn-in iterations and 100,000 samples. To assess convergence, we calculate the Gelman and Rubin potential scale reduction statistic, $\widehat{R}$, for each dimension and ensure that all of the $\widehat{R}$ values are close to 1 [50]. We calculate ESS for each dimension using samples from all of our chains following Gelman et al. [5]. In our experiments, we compute $\widehat{R}$ and ESS using RStan [47]. Since SA-MCMC has a state consisting of $N$ points, we compute ESS for SA-MCMC as $N$ times the effective sample size of the history of the mean of the $N$ points as in Goodman and Weare [19, p. 73-74]. Our experiments and timing are done on a Intel Xeon E5-2640v3 using Julia v0.64 [51], except for NUTS which uses Stan C++.

**Toy 1D examples** We first demonstrate the adaptive nature of SA-MCMC in three different cases in Figure 2. In the first example, the target distribution is $\mathcal{N}(0, 1)$ and our proposal distribution is $\mathcal{N}(-10, 10^2)$. Even though our guess of the mean is far away from the true mean, SA-MCMC is able to quickly hone in on the high-probability region of the target distribution. In the second example, the target is $\mathcal{N}(0, 3^2)$ and our proposal is $\mathcal{N}(-4, 1)$. Though we start with an incorrect mean and an underestimate of the variance, SA-MCMC is able to adapt to the target. In the third example, the target is $\mathcal{N}(0, 1)$ and our proposal is $\mathcal{N}(-5, 1)$. Even when there is little overlap in the densities of the proposal and the target, the proposal is able to move to the target. The adaptivity of SA-MCMC demonstrated here enables tuning-free MCMC, as SA-MCMC can quickly transition from its initial state of $N$ points to an empirical representation of the target distribution during the burn-in phase.

**Bayesian linear regression** We consider a Bayesian linear regression model where the regression parameters have i.i.d. Laplace priors. To study the adaptivity of SA-MCMC, we generate a synthetic dataset where the posterior standard deviation of the regression parameters varies. The true regression parameters $\beta$ are sampled from i.i.d. Laplace$(0, 1)$ priors. For the feature matrix $\mathbf{X}$, each entry of

Table 1: Comparison of ESS/second for Bayesian logistic regression on (***top***) 11-dim MNIST 7s vs 9s using 10 features computed with PCA (***bottom***) 7-dim adult census income

|  | MH | MTM | AM (diag) | AM (full) | SA (diag) | SA (full) | NUTS |
|---|---|---|---|---|---|---|---|
| min(ESS)/s | 13 | 5 | 17 | 37 | 23 | **278** | 54 |
| median(ESS)/s | 21 | 9 | 23 | 38 | 52 | **290** | 105 |
| s/chain | 733 | 3651 | 734 | 742 | 782 | 1112 | 1160 |
| Hyperparameters | $q$=.02 | $q$=.02 | $q$=.02 | $q$=.02 | $q_0$=1 | $q_0$=1 | Stan |
|  |  | $M$=3 | $s$=.6 | $s$=.7 | $N$=40 | $N$=150 |  |
| Acceptance rate | 23% | 48% | 24% | 26% | 75% | 98.9% | — |
| min(ESS)/s | 1.4 | 0.6 | 13 | 16 | 67 | **151** | 40 |
| median(ESS)/s | 17 | 7 | 15 | 17 | 89 | **158** | 49 |
| s/chain | 2198 | 10951 | 2205 | 2217 | 2283 | 2509 | 2989 |
| Hyperparameters | $q$=.016 | $q$=.016 | $q$=.016 | $q$=.016 | $q_0$=1 | $q_0$=1 | Stan |
|  |  | $M$=3 | $s$=.8 | $s$=.85 | $N$=40 | $N$=150 |  |
| Acceptance rate | 26% | 52% | 21% | 24% | 89% | 99.2% | — |

column $j \geq 1$ is sampled i.i.d. from $\mathcal{N}(0, (j+1)^2/4)$. The dependent variables are generated with a high noise level as $\mathbf{y} \sim \mathcal{N}(\mathbf{X}\beta + \beta_0, 10^2)$. For our experiment, we consider 10 regression parameters and a dataset of 10,000 points with a 80%/20% train/test split.

The ESS/second for each regression parameter is presented in Figure 3 (***left***) for each MCMC method. We use SA and AM with diagonal covariance matrices for this experiment. The hyperparameters are (MH) $q$=.03; (MTM) $q$=.03, $M$=3; (AM) $q$=.03, $s$=.7; (SA) $q_0$=1, $N$=40. SA-MCMC achieves very high ESS/second as the SA multivariate Gaussian proposal distribution adapts within its parametric family to match the posterior standard deviation of each regression parameter, as shown in Figure 3 (***right***). For this reason, the ESS of SA is nearly constant across the regression parameters. AM and NUTS are also able to adapt for this problem. Since MH and MTM only use a single scale parameter for the proposal distribution and cannot adapt, MH and MTM are very inefficient in sampling certain coordinates. When comparing min(ESS)/second, MH's is 62, MTM's is 25, AM's is 387, NUTS's is 365, and SA's is 2329. Under this metric, SA is 6x more efficient than AM, 6.4x than NUTS, 38x than MH, and 94x than MTM. The average running time in seconds for each chain is (MH) 66; (MTM) 341; (AM) 70; (NUTS) 493; (SA) 114. Finally, we emphasize that no tuning is required for SA since a Gaussian initialization distribution with standard deviation of 1 suffices.

**Bayesian logistic regression** We consider a Bayesian logistic regression model for binary classification where the prior on the regression coefficients is Gaussian. For our experiments, we use a standard multivariate Gaussian as the prior. We first present results on two large-scale, real-world datasets: classifying digits 7 vs. 9 on the MNIST dataset, and predicting whether an adult's income exceeds $50K/year based on the census income dataset from the UCI repository [52]. The MNIST training set consists of 12,214 images, and after scaling the pixel values to the range $[0, 1]$, we reduce the dimensionality of the image from 784 to 10 using PCA similar to Korattikara et al. [53]. The resulting classification accuracy is around 93%. The census income training set has 32,561 data points, and we use 6 continuous features as predictors (we exclude fnlwgt and include gender). We standardize each feature in the feature matrix to zero mean and unit variance. Visualizations of the posterior distributions are presented in Appendix 7.

The ESS/second results, as well as hyperparameters and acceptance rates, for each MCMC method are presented in Table 1. Overall, the high acceptance rates of 98.9% and 99.2% for SA-MCMC using the full covariance matrix indicate that the posterior distributions are approximated well by Gaussian distributions that can be captured by the adaptive proposal family, leading to high ESS/s for SA-MCMC. For the MNIST dataset, when comparing min(ESS)/second, we see that SA (full) is 5.2x more efficient than NUTS, 7.6x than AM (full), 21x than MH, and 52x than MTM. Since a few dimensions of the posterior are highly correlated, using a full covariance matrix for AM and SA improves ESS. NUTS is adversely affected by the high correlation, and its min(ESS) is around half of its median(ESS). For the census income dataset, when comparing min(ESS)/second, we see that SA (full) is 3.8x more efficient than NUTS, 9.4x than AM (full), 106x than MH, and 263x than MTM. MH and MTM are extremely inefficient in this case because these 2 algorithms are non-adaptive

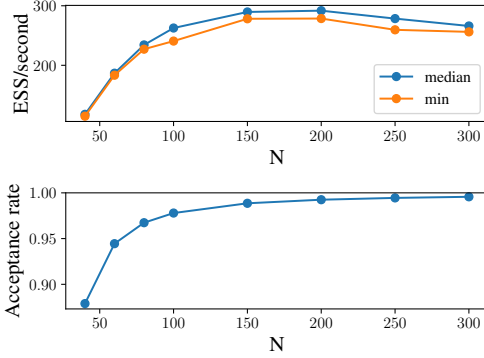

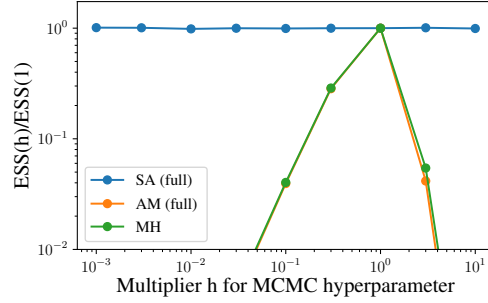

Figure 4: Plot of ESS/s and acceptance rate for SA-MCMC (full) versus $N$ on MNIST.

Figure 5: Impact of MCMC hyperparameter on ESS for MNIST. The ratio ESS(h)/ESS(1) measures the drop in ESS using $0.02h$ for $q$ in MH, $0.7h$ for $s$ in AM, and $1h$ for $q_0$ in SA.

and one of the regression coefficients has a posterior standard deviation of 0.072 while the other 6 regression coefficients have posterior standard deviations of 0.013-0.020. Thus, we see a real-world example of the scenario we presented with Bayesian linear regression.

In Figure 4, we plot ESS/s and acceptance rate for SA-MCMC (full) as a function of $N$ for the MNIST dataset. Note that the acceptance rate approaches 1 as $N$ increases. The ESS/s also increases as we increase $N$ from 40 to 150 with a similar curvature as the acceptance rate plot. Past $N = 200$, the ESS/s starts to slowly decline. In Figure 5, we study the impact of MCMC hyperparameter tuning on ESS for the MNIST dataset. We define ESS(1) to be the median ESS using the optimal hyperparameter in Table 1: 0.02 for $q$ in MH, 0.7 for $s$ in AM (full), and 1 for $q_0$ in SA (full). We define ESS(h) to be the median ESS using the hyperparameter $0.02h$ for $q$ in MH, $0.7h$ for $s$ in AM (full), and $1h$ for $q_0$ in SA (full) and plot the ratio ESS(h)/ESS(1) as we vary $h$. In this experiment, we use $N = 150$ and 500k burn-in iterations followed by 1 million estimation iterations. For any value of $q_0$ from $10^{-3}$ to $10^1$, SA-MCMC can adapt perfectly to the target distribution during the burn-in phase and maintains optimal ESS. In contrast, both AM and MH suffer from suboptimal hyperparameters with ESS dropping significantly. In Appendix 8, we present results for SA-MCMC and NUTS on MNIST across a range of dimensions. When comparing minimum ESS/second, we note that SA-MCMC (full) outperforms NUTS up to dimension 50 on MNIST.

Finally, we present results on two higher-dimensional, large-scale datasets: predicting forest cover type from cartographic variables using the covtype.binary dataset,[1] and distinguishing electron neutrinos (signal) from muon neutrinos (background) based on the MiniBooNE dataset from the UCI repository [52]. The covtype dataset has a total of 581,012 data points, and we use a 80% training and 20% test split. There are 54 features in total, with 10 real-valued features and 44 binary features. The MiniBooNE dataset has 130,065 data points and 50 real-valued features. For MiniBooNE, we normalize each feature to zero mean and unit variance. The covtype and MiniBooNE datasets lead to extremely challenging sampling problems. The condition number of the posterior covariance matrix is around 340,000 for covtype and 140,000 for MiniBooNE.

For this experiment, we first run Newton's method to obtain a point estimate of the posterior mode and then initialize MH, AM (full), and SA (full) around this point estimate. Specifically, if we let $\tilde{\theta}$ be the point estimate, then we initialize $\theta^{(0)} \sim q_{0,\text{MH}}(\cdot) = \mathcal{N}\left(\tilde{\theta}, \sigma^2_{q,\text{MH}}\mathbb{I}\right)$ for MH and $q_{0,\text{SA}}(\cdot) = \mathcal{N}\left(\tilde{\theta}, \sigma^2_{q0,\text{SA}}\mathbb{I}\right)$ for SA. MH with an isotropic normal distribution as the proposal distribution is not able to sample all of the dimensions effectively with ESS and R_hat detecting non-convergence in several dimensions. Since NUTS with the default options in Stan does not work well for this problem, we run NUTS with a dense mass matrix instead of a diagonal mass matrix. The ESS/second results are presented in Table 2. When comparing min(ESS)/s, SA outperforms AM by 31x on covertype and 11x on MiniBoonE and outperforms NUTS by 24x on covertype and 147x on MiniBoonE. Thus, SA samples effectively from this high-dimensional, challenging posterior distribution without requiring any tuning of the initialization distribution. While we use 500 burn-in

Table 2: Comparison of ESS/second for Bayesian logistic regression on (**left**) 55-dim cover type (**right**) 51-dim MiniBooNE between AM (full), SA (full), and NUTS with a dense mass matrix.

| | Cover type | | | MiniBooNE | | |
|---|---|---|---|---|---|---|
| | AM | SA | NUTS | AM | SA | NUTS |
| min(ESS)/s | 0.075 | **2.34** | 0.099 | 0.31 | **3.35** | 0.023 |
| median(ESS)/s | 0.078 | **2.81** | 0.114 | 0.38 | **6.59** | 0.039 |
| s/chain | 52,469 | 65,537 | 25,143 | 28,178 | 26,627 | 33,584 |
| s/chain (burn-in) | 4,770 | 5,958 | 16,980 | 1,342 | 2,421 | 19,051 |
| s/chain (estimation) | 47,699 | 59,579 | 8,163 | 26,836 | 24,206 | 14,533 |
| # iter. (burn-in) | 100,000 | 100,000 | 500 | 100,000 | 100,000 | 500 |
| # iter. (estimation) | 1,000,000 | 1,000,000 | 2,000 | 2,000,000 | 1,000,000 | 2,000 |
| Hyperparameters | $q$=.004 $s$=.32 | $q_0$=1 $N$=1,000 | Stan (dense) | $q$=.007 $s$=.33 | $q_0$=1 $N$=1000 | Stan (dense) |
| Acceptance rate | 25.1% | 99.3% | — | 25.7% | 90.5% | — |

and 2,000 estimation iterations for NUTS, the running time of NUTS in the burn-in phase is larger than in the estimation phase due to the number of likelihood evaluations. Note that our ESS/second calculation is based on the total running time of the algorithm, including burn-in. We note that it is possible that further tuning or other techniques could improve the performance of NUTS.

## 5 Discussion

Our experimental results demonstrate the strong empirical performance of SA-MCMC with zero tuning compared to MH, MTM, AM, and NUTS with extensive tuning on Bayesian linear regression and Bayesian logistic regression. SA-MCMC achieves this by maintaining a state of $N$ points and using an adaptive proposal distribution $q(\cdot|\mu(S), \Sigma(S))$ depending on the current state. The SA-MCMC substitution procedure for the $N$ points guarantees that the proposal distribution adapts within its parametric family to best approximate the target distribution. For example, when using a Gaussian family of proposal distributions, SA-MCMC is well-suited for posterior inference tasks where the posterior distribution can be approximated well by a Gaussian distribution. In these cases, SA-MCMC is very efficient as the draws from the proposal distribution approximate draws from the target distribution. While we focused on proposal families of the form $q(\cdot|\mu(S), \Sigma(S))$ in this paper, more generally, our method can be extended to proposals of the form $q(\cdot|\gamma(S))$ where $\gamma(S) = N^{-1} \sum_{\theta \in S} \gamma(\theta)$ (as proved in Theorem 2) to tackle other problems. Future extensions of this work include using a family of mixture distributions as the proposal family and learning the optimal mixture distribution (within a given family) [54, 55] and combining SA-MCMC updates with other MCMC updates, such as with NKC in the Parallel Metropolis-Hastings Coupler [56].

The computational complexity per iteration of SA-MCMC is one likelihood evaluation and the computation of the substitution probabilities in time $O(Nd)$ with a a diagonal covariance matrix or time $O(Nd^2)$ with the full covariance matrix where $d$ is the dimension. The computational complexity per iteration of MH and AM is one likelihood evaluation plus $O(d)$ with a diagonal covariance matrix or $O(d^2)$ with the full covariance matrix. The computational complexity per iteration of MTM with $M$ tries is $2M - 1$ likelihood evaluations plus $O(d)$ or $O(d^2)$.

While the adaptivity of AM is based on the entire history of past samples, the adaptivity of SA-MCMC is based on the current state of $N$ points which offers theoretical and experimental advantages. For SA-MCMC, the Markovian property of the chain and the reversibility of the chain are preserved, and standard MCMC convergence theory can be applied. With AM, the first stage of AM is MH, so the MH proposal distribution during the non-adaptive phase still has to be tuned. Using a sequential substitution framework, SA-MCMC is a principled adaptive MCMC method that only requires specifying an initialization distribution. In many cases, the initialization distribution for SA-MCMC can be chosen *a priori*, thereby automating the entire sampling procedure with no tuning required. Experimental results demonstrate the fast adaptation and effective sampling of SA-MCMC.

**Acknowledgments**

I would like to thank my advisor, Professor Tze Leung Lai, for introducing me to this research area and for supporting me throughout this project. I would like to thank Tze Leung Lai and Hock Peng Chan for providing a working paper with the algorithm and some theoretical derivations including the proof for uniform ergodicity. Finally, I would like to thank the anonymous reviewers for their valuable feedback.

## Footnotes

[1] `https://www.csie.ntu.edu.tw/~cjlin/libsvmtools/datasets/binary.html`

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
