[Supplementary Material · supplement.pdf]

# Supplementary Material for Sample Adaptive MCMC

**Michael H. Zhu**
Department of Computer Science
Stanford University
Stanford, CA 94305
mhzhu@cs.stanford.edu

## Appendix 1: Proof of detailed balance

*Proof.* We prove that the SA-MCMC Markov chain satisfies the detailed balance condition. Let $S = (\theta_1, \theta_2, \ldots, \theta_N)$, and let $\mathbb{T}$ denote the transition kernel of the chain. To show the detailed balance condition, we need to show that $\pi^{\otimes N}(S)\mathbb{T}(S'|S) = \pi^{\otimes N}(S')\mathbb{T}(S|S')$ for $S' \neq S$. Consider the case where $S' = (\theta_1', \theta_2, \ldots, \theta_N)$ for $\theta_1' \neq \theta_1$:

$$\pi^{\otimes N}(S)\mathbb{T}(S'|S) = \frac{\left[\pi(\theta_1)\prod_{n=2}^N \pi(\theta_n)\right][q(\theta_1'|\mu(S),\Sigma(S))]\frac{q(\theta_1|\mu(S_{-1}),\Sigma(S_{-1}))}{p(\theta_1)}}{\frac{q(\theta_1|\mu(S_{-1}),\Sigma(S_{-1}))}{p(\theta_1)} + \frac{q(\theta_1'|\mu(S),\Sigma(S))}{p(\theta_1')} + \sum_{j=2}^N \frac{q(\theta_j|\mu(S_{-j}),\Sigma(S_{-j}))}{p(\theta_j)}}$$

$$\pi^{\otimes N}(S')\mathbb{T}(S|S') = \frac{\left[\pi(\theta_1')\prod_{n=2}^N \pi(\theta_n)\right][q(\theta_1|\mu(S'),\Sigma(S'))]\frac{q(\theta_1'|\mu(S_{-1}'),\Sigma(S_{-1}'))}{p(\theta_1')}}{\frac{q(\theta_1'|\mu(S_{-1}'),\Sigma(S_{-1}'))}{p(\theta_1')} + \frac{q(\theta_1|\mu(S'),\Sigma(S'))}{p(\theta_1)} + \sum_{j=2}^N \frac{q(\theta_j|\mu(S_{-j}'),\Sigma(S_{-j}'))}{p(\theta_j)}}$$

Since $S' = S_{-1}$ and $S_{-1}' = S$ and $\mu(S_{-j}') = \mu(S_{-j})$ and $\Sigma(S_{-j}') = \Sigma(S_{-j})$ for $j = 2, ..., N$, the detailed balance condition is satisfied for $S \leftrightarrow S'$. By symmetry, the same argument holds when any one of the $\theta_n$'s is updated. Since at most one $\theta_n$ can be updated in one iteration, we are done. $\square$

## Appendix 2: Proof of ergodicity

We first restate a theorem from Athreya et al. (1996) that we will use to prove Theorem 1. Our statement of the theorem is also based on Robert and Casella (2004, p. 263).

**Theorem** (Athreya et al., 1996). *Suppose that the time-homogenous Markov Chain $\{X^{(k)}\}$ has invariant probability density $p$. Denote the conditional density of $X^{(k)}$ given $X^{(0)}$ by $f_k(\cdot|\cdot)$. Let $g(x)$ be any function satisfying $\int |g(x)|\, p(x)dx < \infty$. Suppose that there is a set $A$ with $\int_A p(x)dx > 0$ that satisfies the following properties:*

*(B1) $\sum_{k=1}^\infty \int_A f_k(x|x_0)dx > 0$ for $[p]$-almost all $x_0$, and*

*(B2) $\gcd\{m\colon \inf_{x,y \in A} f_m(y|x) > 0\} = 1$.*

*Then the Markov chain is ergodic, and*

*(1) $\lim_{K \to \infty} \sup_C \left|\int_C f_K(x|x_0)dx - \int_C p(x)dx\right| = 0$ for $[p]$-almost all $x_0$, and*

*(2) $\mathbb{P}_{x_0}\left[\lim_{K \to \infty}\frac{1}{K}\sum_{k=1}^K g(X^{(k)}) = \int g(x)p(x)dx\right] = 1$ for $[p]$-almost all $x_0$.*

*Proof of Theorem 1.* From Proposition 1, we know that $\{S^{(k)}\}$ has invariant density $\pi^{\otimes N}$. We will first show that assumption (B1) of the Theorem of Athreya et al. (1996) is satisfied for any set

$A \subseteq \mathcal{E}^{\otimes N}$ with $\int_A \pi^{\otimes N}(s)ds > 0$. Consider any $s^{(0)} = (\theta_1^{(0)}, \ldots, \theta_N^{(0)})$ with positive probability $\pi^{\otimes N}(s^{(0)}) > 0$ and not all $\theta_i^{(0)}$ identical to each other. Consider any $s = (\theta_1, \ldots, \theta_N) \in A$ with not all $\theta_i$ identical to each other and $\theta_i \neq \theta_j^{(0)}$ for any $i, j$. $s$ can be reached from $s^{(0)}$ in $N$ steps by proposing $\theta_i$ and replacing $\theta_i^{(0)}$ with $\theta_i$ in step $i$. Since the proposal densities and the target densities are positive by assumption, the substitution probabilities in each step are positive, so $\int_A f_N(s|s_0)ds > 0$.

We now show that assumption (B2) is satisfied. Without loss of generality, we can choose our coordinate system so that there exists some $\delta > 0$ such that $[-6\delta, 6\delta]^d \subseteq \mathcal{E}$. Consider the case where $N \geq 4$ is even. Let $A$ be the set $[-4\delta, -3\delta]^{dN/2} \times [3\delta, 4\delta]^{dN/2}$. For any $s \in A$, our construction ensures that $\theta_1[j], \ldots, \theta_{N/2}[j] \in [-4\delta, -3\delta]$ and $\theta_{N/2+1}[j], \ldots, \theta_N[j] \in [3\delta, 4\delta]$ for any dimension $j \in 1, \ldots, d$. To bound all of the substitution probabilities during the substitution procedure for our proof, we will also need to consider the cases where there are $(\frac{N}{2} + 1) \theta_n[j]$'s in $[-4\delta, -3\delta]$ and $(\frac{N}{2} - 1) \theta_n[j]$'s in $[-4\delta, -3\delta]$ with the remaining $\theta_n[j]$'s in $[3\delta, 4\delta]$. In any of these 3 cases, we have that $|\mu_j| \leq \frac{1}{N} \left(4\delta(\frac{N}{2} + 1) - 3\delta(\frac{N}{2} - 1)\right) = \frac{\delta}{2} + \frac{7\delta}{N} \leq \frac{9\delta}{4}$ for $N \geq 4$ and $\frac{3\delta}{4} \leq \sigma_j \leq 8\delta$. For any $\theta' \in [-4\delta, 4\delta]^d$, we have that $|\theta'_j - \mu_j| \leq 8\delta$, so by assumption (A2), there exist constants $\epsilon_1, \epsilon_2 > 0$ such that $\epsilon_1 < q(\theta' \mid \mu, \text{diag}(\sigma^2)) < \epsilon_2$. By assumption (A1), there exist constants $\epsilon_3, \epsilon_4 > 0$ such that $\epsilon_3 < \pi(\theta) < \epsilon_4$ for all $\theta$ in $[-4\delta, 4\delta]^d$.

We are now ready to prove that $\inf_{s^{(0)}, s \in A} f_N(s|s^{(0)}) > 0$. Let $s^{(0)} = (\theta_1^{(0)}, \ldots, \theta_N^{(0)}) \in A$ and $s = (\theta_1, \ldots, \theta_N) \in A$. Since $s$ can be reached from $s^{(0)}$ in $N$ steps by proposing $\theta_i$ and replacing $\theta_i^{(0)}$ with $\theta_i$ in step $i$, we need only show that this $N$-step transition probability is bounded from below. Making use of the bounds proved earlier, the probability of proposing $\theta_i$ and replacing $\theta_i^{(0)}$ with $\theta_i$ in step $i$ is at least $\frac{\epsilon_1^2 \epsilon_3}{(N+1)\epsilon_4 \epsilon_2}$. Combining the $N$ steps proves that $\inf_{s^{(0)}, s \in A} f_N(s|s^{(0)}) \geq \left(\frac{\epsilon_1^2 \epsilon_3}{(N+1)\epsilon_4 \epsilon_2}\right)^N$. The same argument can be used to prove $\inf_{s^{(0)}, s \in A} f_m(s|s^{(0)}) > 0$ for any $m \geq N$ which establishes (B2).

Result (1) follows immediately from result (1) of the Theorem of Athreya et al. (1996). To conclude result (2), note that we can define $g(s) = \frac{1}{N} \sum_{n=1}^N h(\theta_n)$, and $\int |h(\theta)| \pi(\theta)d\theta < \infty$ implies $\int |g(s)| \pi^{\otimes N}(s)ds \leq \frac{1}{N} \sum_{n=1}^N \int |h(\theta_n)| \pi^{\otimes N}(s)ds = \int |h(\theta)| \pi(\theta)d\theta < \infty$, so result (2) follows from result (2) of the Theorem of Athreya et al. (1996).

In the case where $N \geq 5$ is odd, we can let $A$ be the set $[-4\delta, -3\delta]^{d(N-1)/2} \times [3\delta, 4\delta]^{d(N+1)/2}$. Following the same argument above, we have that $|\mu_j| \leq \frac{1}{N} \left(4\delta(\frac{N+1}{2} + 1) - 3\delta(\frac{N-1}{2} - 1)\right) = \frac{\delta}{2} + \frac{21\delta}{2N} \leq \frac{26\delta}{10}$ for $N \geq 5$ and $\frac{4\delta}{10} \leq \sigma_j \leq 8\delta$, and the rest of the proof follows.

In the case where $N = 3$, we can let $A$ be the set $\mathcal{I}_1 \times \mathcal{I}_2 \times \mathcal{I}_3 = [-6\delta, -5\delta]^d \times [-\delta, \delta]^d \times [5\delta, 6\delta]^d$. We now analyze the different cases. When we have one $\theta_n$ in each of $\mathcal{I}_1, \mathcal{I}_2, \mathcal{I}_3$, then $|\mu_j| \leq \frac{1}{3}(-5\delta + \delta + 6\delta) = \frac{2\delta}{3}$ and $\sigma_j^2 \geq \frac{2}{3}(4\delta)^2$. When we have two $\theta_n$'s in $\mathcal{I}_1$ and one $\theta_n$ in either $\mathcal{I}_2$ or $\mathcal{I}_3$, then $\mu_j \leq \frac{1}{3}(-10\delta + 6\delta) = \frac{-4\delta}{3}$ and $\sigma_j^2 \geq \frac{1}{3}(\frac{\delta}{3})^2$. Finally, when we have two $\theta_n$'s in $\mathcal{I}_2$ and one $\theta_n$ in $\mathcal{I}_3$, then $\mu_j \leq \frac{1}{3}(2\delta + 6\delta) = \frac{8\delta}{3}$ and $\sigma_j^2 \geq \frac{1}{3}(2\delta)^2$. In any case, we have the upper bound $\sigma_j^2 \leq (12\delta)^2$. For any $\theta' \in [-6\delta, 6\delta]^d$, we have that $|\theta'_j - \mu_j| \leq 12\delta$, and the rest of the proof follows. $\square$

## Appendix 3: Proof of uniform ergodicity

*Proof.* For notational convenience, let $p$ denote the target density. Let $\Psi$ be the linear operator on $\mathcal{M}[=\mathcal{M}(\Theta^N)]$, the space of finite signed-measures on $\Theta^N$, such that $\Psi(\delta_{\boldsymbol{\theta}})$ is the probability measure of $(\theta_1^{(k+1)}, \ldots, \theta_N^{(k+1)})$, conditioned on $(\theta_1^{(k)}, \ldots, \theta_N^{(k)}) = \boldsymbol{\theta}$.

Let $q_0(\theta) = \inf_{\gamma \in \Gamma} q(\theta|\gamma)$. Since $\lambda(\theta|\gamma)$ is bounded by assumption, $q_0 \geq ap$ (i.e. $q_0 - ap$ is a non-negative measure). We will first show that

$$\Psi^N(\delta_{\boldsymbol{\theta}}) \geq C_0 q_0^N \geq C p^N, \tag{1}$$

where $C_0 = N!(\frac{a}{(N+1)b})^N$ and $C = C_0 a^N$. This is because $\Psi^N$ includes the special case of substitutions of each $\theta_n$ ($1 \leq n \leq N$) in $N$ consecutive operations. For a particular sequence of substitutions, say $\theta_1$ followed by $\theta_2$ up to $\theta_N$, the probability of this occurring is at least $(\frac{a}{(N+1)b})^N$. We multiply by $N!$ to take into account all possible orders of substitutions.

We shall prove by induction that for $\ell = 0, 1, 2, \ldots$,

$$\Psi^{\ell N}(\delta_{\boldsymbol{\theta}}) \geq [1 - (1-C)^{\ell}]p^N. \tag{2}$$

Indeed (2) is trivial for $\ell = 0$. Now suppose (2) holds for some $\ell$. We can rewrite

$$\Psi^{(\ell+1)N}(\delta_{\boldsymbol{\theta}}) = \Psi^N([1 - (1-C)^{\ell}]p^N) + \Psi^{(\ell+1)N}(\delta_{\boldsymbol{\theta}}) - \Psi^N([1 - (1-C)^{\ell}]p^N)$$
$$= \Psi^N([1 - (1-C)^{\ell}]p^N) + (1-C)^{\ell}\Psi^N(\nu^*)$$

where $\nu^* = (1-C)^{-\ell}\{\Psi^{\ell N}(\delta_{\boldsymbol{\theta}}) - [1 - (1-C)^{\ell}]p^N\}$ is a probability measure. Since $\Psi(p^N) = p^N$ and $\Psi^N(\nu^*) \geq Cp^N$ by (1), hence

$$\Psi^{(\ell+1)N}(\delta_{\boldsymbol{\theta}}) \geq [1 - (1-C)^{\ell} + C(1-C)^{\ell}]p^N,$$

so (2) holds for $\ell + 1$, and the induction is complete.

By (2), for $k \geq 1$ and $\ell = \lfloor k/N \rfloor$,

$$\|f_k - p^N\|_{\mathrm{TV}} \leq \|f_{\ell N} - p^N\|_{\mathrm{TV}} \leq 2(1-C)^{\ell} = 2(1-C)^{\lfloor k/N \rfloor}. \tag{3}$$

The first inequality in (3) follows from $\|\Psi(\nu)\|_{\mathrm{TV}} \leq \|\nu\|_{\mathrm{TV}}$ for $\nu \in \mathcal{M}$. $\qquad\square$

## Appendix 4: Experiments with t-distributions

Table 1: Comparison of minimum ESS/iteration for SA-MCMC (diag) with $N = 50$ when using the proposal distribution specified by the row to sample from the target distribution specified by the column in a 10 dimensional space. The proposal family for the t-distribution with $\nu$ degrees of freedom is given by $q(\cdot|\mu(S), \Sigma(S)) = \mathcal{T}_{\nu}(\cdot|\mu(S), \frac{\nu-2}{\nu}\mathrm{diag}(\Sigma(S)))$ where $\nu$ is fixed. 100 chains with 100k burn-in iterations and 1 million estimation iterations each were used to compute ESS. "—" indicates non-convergence as detected by ESS and R_hat.

|              | $T(\nu=3)$ | $T(\nu=5)$ | $T(\nu=10)$ | $T(\nu=50)$ | $T(\nu=100)$ | Normal |
|--------------|------------|------------|-------------|-------------|--------------|--------|
| $T(\nu=3)$   | 0.257      | 0.248      | 0.189       | 0.137       | 0.131        | 0.125  |
| $T(\nu=5)$   | 0.067      | 0.326      | 0.299       | 0.233       | 0.223        | 0.212  |
| $T(\nu=10)$  | —          | —          | 0.345       | 0.310       | 0.297        | 0.287  |
| $T(\nu=50)$  | —          | —          | —           | 0.350       | 0.350        | 0.347  |
| $T(\nu=100)$ | —          | —          | —           | 0.339       | 0.346        | 0.349  |
| Normal       | —          | —          | —           | —           | 0.340        | 0.347  |

# Appendix 5: Discussion of Independent Metropolis-Hastings and Adaptive Independent Metropolis-Hastings

---

**Algorithm 1** Independent Metropolis-Hastings

---

1: Generate an initial state $x_0$ from the density $q_0(\cdot)$
2: **for** $t = 1$ **to** $\kappa + K$ **do**
3:     Generate a state $z$ from the proposal $q(\cdot)$
4:     Calculate the acceptance probability

$$\alpha(z, x_{t-1}) = \min\left\{1, \frac{\pi(z)q(x_{t-1})}{\pi(x_{t-1})q(z)}\right\}.$$

5:     If it is accepted, set $x_t = z$. Otherwise, set $x_t = x_{t-1}$.
6: **end for**

---

---

**Algorithm 2** Adaptive Independent Metropolis-Hastings (with no local steps)

---

1: Initialize $\widetilde{\mathbf{y}}^{(0)} \leftarrow \varnothing$
2: Generate an initial state $x_0$ from the density $q_0(\cdot)$
3: **for** $t = 1$ **to** $\kappa + K$ **do**
4:     Generate a state $z$ from the proposal $q_t(\cdot \mid \widetilde{\mathbf{y}}^{(t-1)})$.
5:     Calculate the acceptance probability

$$\alpha(z, x_{t-1}, \widetilde{\mathbf{y}}^{(t-1)}) = \min\left\{1, \frac{\pi(z)q_t(x_{t-1} \mid \widetilde{\mathbf{y}}^{(t-1)})}{\pi(x_{t-1})q_t(z \mid \widetilde{\mathbf{y}}^{(t-1)})}\right\}.$$

6:     If it is accepted, set $x_t = z$ and $\widetilde{\mathbf{y}}^{(t)} = \widetilde{\mathbf{y}}^{(t-1)} \cup \{x_{t-1}\}$. Otherwise, set $x_t = x_{t-1}$ and $\widetilde{\mathbf{y}}^{(t)} = \widetilde{\mathbf{y}}^{(t-1)} \cup \{z\}$.
7: **end for**

---

The Independent Metropolis-Hastings algorithm uses a fixed proposal distribution, and the proposed point is drawn independently of the current state of the Markov chain. The efficiency of the IMH sampler depends crucially on how well the proposal distribution matches the target distribution, and the IMH algorithm requires that the proposal distribution be specified by the user. Robert and Casella (2004, p. 284) make the following remark: "A final note about independent Metropolis-Hastings algorithms is that they cannot be omniscient: there are settings where an independent proposal does not work well because of the complexity of the target distribution. Since the main purpose of MCMC algorithms is to provide a crude but easy simulation technique, it is difficult to imagine spending a long time on the design of the proposal distribution. This is specially pertinent in high-dimensional models where the capture of the main features of the target distribution is most often impossible. There is therefore a limitation of the independent proposal, which can be perceived as a *global* proposal, and a need to use more *local* proposals that are not so sensitive to the target distribution."

Visualizations of the marginal posterior distribution for the Bayesian logistic regression examples are shown in Figure 1 and Figure 2. Note that the marginal posterior distribution in each dimension is sharply peaked with the peaks concentrated far away from one another, so that the design of a global proposal is difficult. In addition, the posterior parameters can be correlated with each other, as shown in Figure 3 for the MNIST example, further complicating the design of a global proposal distribution that matches the target distribution. Given these challenges in high-dimensional spaces, we view adaptivity of the proposal distribution as crucial for effective sampling.

The Adaptive Independent Metropolis-Hastings algorithm allows the proposal distribution to depend on a history vector, so the algorithm can adapt based on the past samples. The specific form of the algorithm is chosen to preserve the invariant distribution. However, the adaptivity can be limited and crucially depends on the design of a good initial proposal distribution at the start of the algorithm. Note that if the initial proposal distribution is bad so that very few proposed points are accepted, then the history vector is appended with all of the rejected points. Since the history vector consists mostly of rejected points sampled from the initial proposal distribution, then the adaptivity that can occur

is limited. This is in sharp contrast to SA-MCMC which can adapt even when the initial proposal distribution is a bad match for the target distribution.

## Appendix 6: Optimized implementation

We now develop an optimized implementation that is numerically stable and can be vectorized. For numerical stability, all calculations with probabilities are done in the log domain. To avoid repeated evaluation of $p(\theta)$, we cache $p(\theta)$ so that $p(\theta)$ only needs to be evaluated once for each new point $\theta$. Most of the computation required for SA-MCMC occurs in the calculation of $q(\theta_n | \mu(S_{-n}), \Sigma(S_{-n}))$ for $n = 1, \ldots, N+1$, and we now present a more efficient implementation.

### Diagonal covariance

Recall that $S_{-n}$ is ($S$ with $\theta_n$ replaced by $\theta_{N+1}$). The idea is to use the identity $\mathrm{Var}\,[X] = \mathrm{E}\,[X^2] - \mathrm{E}\,[X]^2 = \frac{1}{n}\sum_{i=1}^{n} x_i^2 - \left(\frac{1}{n}\sum_{i=1}^{n} x_i\right)^2$ so that we can incrementally compute the variance $\sigma^2(S_{-n})$ without iterating over $N$ points for each $n$. However, the naive use of this formula can be numerically unstable in certain cases. Instead, we will utilize the translation invariance property of the variance by translating all points by $\mu(S)$ in all of the calculations. For any $n$, we have $\mu_*(S_{-n}) = \frac{1}{N}\left(\theta_{N+1} - \theta_n\right)$. Let sqr denote the element-wise square function. Given $s_* = \sum_{n=1}^{N+1} \mathrm{sqr}\,(\theta_n - \mu(S))$, we can compute $\sigma^2(S_{-n}) = \sigma_*^2(S_{-n}) = \frac{1}{N}\left(s_* - \mathrm{sqr}\,(\theta_n - \mu(S))\right) - \mathrm{sqr}\,(\mu_*(S_{-n}))$. All of these calculations as well as the calculation of the log proposal densities $\log q(\theta_n | \mu(S_{-n}), \mathrm{diag}(\Sigma(S_{-n})))$ can be vectorized across $n$, giving us an efficient CPU implementation that can scale to reasonably sized problems.

### Full covariance

We apply a similar technique as in the diagonal case to the covariance formula. Let $\Theta = [\theta_1, \ldots, \theta_N]$ be a $d \times N$ matrix where the $n^{th}$ column is the vector $\theta_n$. Let $\mu = \mu(S)$ be the mean of the $N$ points. The unscaled covariance matrix $\Sigma(S)$ can be calculated as $\Sigma(S) = (\Theta - \mu 1^T)(\Theta - \mu 1^T)^T$ where $1$ is the vector of ones. Now consider $S_{-n}$ which is ($S$ with $\theta_n$ replaced by $\theta_{N+1}$). We can compute $\Sigma(S_{-n})$ from $\Sigma(S)$ by 3 rank-one updates, corresponding to adding $(\theta_{N+1} - \mu)(\theta_{N+1} - \mu)^T$, subtracting $(\theta_n - \mu)(\theta_n - \mu)^T$, and subtracting $1/N$ times $(\theta_{N+1} - \theta_n)(\theta_{N+1} - \theta_n)^T$. Since the first update above is independent of $n$, we need to do 2 rank-one updates for each $n$.

When the proposal distribution is the Gaussian distribution, the computation can be further accelerated. The proposal density $\log q(\theta_n | \mu(S_{-n}), \Sigma(S_{-n}))$ when $q$ is multivariate Gaussian can be computed directly from the Cholesky factorization of $\Sigma(S_{-n})$ which is more efficient than starting with $\Sigma(S_{-n})$. Using this idea, the procedure above of computing $\Sigma(S_{-n})$ from $\Sigma(S)$ by 2 rank-one updates can be improved. We first compute the Cholesky factorization of $\Sigma(S) + (\theta_{N+1} - \mu)(\theta_{N+1} - \mu)^T$. Then for each $n$, we apply the 2 rank-one updates (specifically downdates because of the negative sign) directly to the Cholesky factorization of $\Sigma(S)$ to get the Cholesky factorization of $\Sigma(S_{-n})$, which we can then use to evaluate the density $\log q(\theta_n | \mu(S_{-n}), \Sigma(S_{-n}))$. The runtime complexity of this procedure is $O(Nd^2)$ for each iteration.

# Appendix 7: Visualizations of the posterior distribution for the Bayesian logistic regression examples

Figure 1: Marginal posterior distribution (blue histogram) for the MNIST logistic regression example. For reference, the density of the standard normal distribution is in green.

Figure 2: Marginal posterior distribution (blue histogram) for the census income logistic regression example. For reference, the density of the standard normal distribution is in green.

Figure 3: Bivariate posterior distribution of dimension 3 vs every other dimension in the MNIST logistic regression example.

Figure 4: Plot of minimum ESS/s for SA-MCMC versus $N$ (solid lines) and plot of minimum ESS/s for NUTS (horizontal dashed lines) for different dimensions (different colors).

Figure 5: Plot of median ESS/s for SA-MCMC versus $N$ (solid lines) and plot of median ESS/s for NUTS (horizontal dashed lines) for different dimensions (different colors).

Figure 6: Plot of acceptance rate for SA-MCMC (full) versus $N$ on MNIST for different dimensions.

Figure 7: Plot of running time for SA-MCMC (full) versus $N$ on MNIST for different dimensions.

## Appendix 8: Results for SA-MCMC and NUTS on MNIST in higher dimensions

To study the performance of SA-MCMC (full) and NUTS across different dimensions, we run the following experiment on the MNIST dataset. We reduce the dimensionality of the image from 784 to 20, 30, 40, and 50 using PCA, resulting in 21, 31, 41, and 51 dimensions total respectively after adding a column of ones. We then run NUTS and SA-MCMC (full) with different values of $N$. For SA-MCMC, we use 200k burn-in iterations followed by 1 million estimation iterations. We use $q_0 = 1$ which is the same initialization distribution that we used in all of our other experiments. Note that SA-MCMC is able to adapt to the target distribution in higher dimensions and for larger values of $N$. The minimum ESS/second results are in Figure 4, and the median ESS/second results are in Figure 5. When comparing minimum ESS/second, we note that SA-MCMC outperforms NUTS up to dimension 50 on MNIST. For SA-MCMC, we plot the acceptance rates and running times versus $N$ in Figure 6 and Figure 7.