[Reviews · NeurIPS 2019]

Reviewer 1



EDIT: After reading the author's rebuttal, I changed my assessment of the paper to an accept. The paper is well written and it does a good job at explaining the intuition behind the proposed algorithm. I appreciated the inclusion of the small dimensional toy example as it illustrates in a simple and clear manner the adaptability property of the algorithm. My main concern with the proposed algorithm is that, in my opinion, it is most suitable for small dimensional problems only. The provided examples further justify my impression given that posterior distribution to sample from is of reduced dimension. Consequently, I'm having a hard time justifying the interest of the ML community with respect to the proposed sampling algorithm considering its perceived limited scope. The authors acknoweledge that Sequential Monte Carlo algorithms partly served as an inspiration for the proposed algorithm, however there is no comparison of the proposed algorithm with SMC algorithms. I feel that such a comparison is warranted given that both approaches use a set of points to construct an empirical estimate of the target distribution. I am curious, and I presume that I'm not the only one, as to how the proposed algorithm stacks up to SMC algorithms in a high dimensional setting. I feel that not enough attention is devoted to the case when we restrict ourselves to a diagonal covariance matrix for the proposal distribution. If one wants to apply the proposed algorithm in a high dimensional setting, then this is the only viable manner to do it. In such a scenario we cannot afford to store in memory the full covariance matrix for the proposal distribution. Furthermore, it would not at all be trivial to sample from the proposal distribution even when it is a gaussian distribution. It would be interesting to see and analyze how the generated samples manage to capture the covariance structure of the target distribution. I will list in the following two small issues that I encountered when reading the paper. The first issue is that there are way too many citations, 62 for an 8 page article. Whenever there are multiple sources for an algorithm, please cite only the most representative one. The second issue is that the text in figures 1 and 2 is hardly visible when the article is printed, it would be really great to increase the size of the text.

Reviewer 2



Originality: The specific algorithm and ergodicity results are novel (however, as pointed out by the authors, it is a special case of the sequential substitution framework proposed in [11, 12]. Quality/clarity: In my view, the manuscript is well written and the methodology is presented in a clear and sufficiently rigorous manner. Post-rebuttal edit: ------------------------------------------------------------------------------- The authors' rebuttal provides further (empirical) support for the idea that the proposed MCMC kernel can work well even in higher dimensions. Furthermore, as already mentioned in my review, the presentation of the methodology in the paper is clear and there is substantial amount of theoretical support. The same cannot be said for most other NeurIPS submissions that I have seen this year. As a result, I am happy to raise my recommendation from 6 to 7.

Reviewer 3



The method is interesting and the paper is well-written. Moreover, it seems technically sound. Perhaps the contribution is a bit incremental and the degree of novelty is not very high. However, the state-of-the-art is well done, the paper is easy to read. I believe that in the literature, there is also a need of papers which focus on a detailed study of the previous literature in order to yield important and required variants, as new algorithms. I have just some suggestions for completing the related references, clarify some points and maybe possible future works. - I believe that your proposal could be itself a mixture of densities. In this sense, you could mix your work with the ideas in Cappe et al, “Adaptive Importance Sampling in general mixture classes”, Statistics and Computing 2008, or better for MCMC to have a look to Luengo and Martino, "Fully Adaptive Gaussian Mixture Metropolis-Hastings Algorithm", IEEE International Conference on Acoustics, Speech, and Signal Processing (ICASSP), Vancouver (Canada), 2013. Please discuss. - Another possible approach for using a mixture is the nice idea in G. R. Warnes, “The Normal Kernel Coupler: An adaptive Markov Chain Monte Carlo method for efficiently sampling from multi-modal distributions,” Technical Report, 2001. That you already consider as a reference. In any case, even without considering the possible use of a mixture, the discussion regarding the Normal Kernel Coupler (NKC) should be extended since in this method there is also the idea of replacing one sample inside a population (one per iteration). In this sense there are several connections with the SA-MCMC method. Please discuss relationships and differences. In this sense, I also suggest to have a look to the recent reference F. Llorente et al, "Parallel Metropolis-Hastings Coupler", IEEE Signal Processing Letters, 2019, which combines OMCMC and NKC. It could be nice to discuss it also as a possible future work (extension) of your paper. These discussions can improve substantially the paper and make it a very complete and nice piece of work. I also suggest to upload your revised work to arXiv and/or ResearchGate to increase its impact.

[Author Response · NeurIPS 2019]

Table 1: Comparison of MSE for Bayesian linear regression in a 1,000 dimensional space with 8,000 training examples

|  | MSE | Total runtime | # Burn-in iters | # Total iters | Hyperparameters | Acceptance rate |
|---|---|---|---|---|---|---|
| MH | 0.273 | 35 mins | 50,000 | 500,000 | $q$=0.0035 | 23% |
| AM (diag) | 0.0267 | 35 mins | 50,000 | 500,000 | $q$=0.0035, $s$=0.05 | 22% |
| AM (diag) | 0.00729 | 70 mins | 100,000 | 1,000,000 | $q$=0.0035, $s$=0.05 | 29% |
| SA (diag) | **0.00174** | **33 mins** | 50,000 | **100,000** | $q_0$=1, $N$=500 | 45% |

We thank the reviewers very much for their time and valuable feedback. We will incorporate all of the suggestions when revising our paper, and we will post our revised work to arXiv to increase its impact and audience.

**Reviewer 1:** For the experiments in our paper, we focused on dimensions which we think are commonly used to understand, compare and benchmark MCMC methods. We note that our MCMC method outperforms NUTS, the state-of-the-art gradient-based MCMC sampler, on the MNIST dataset in dimensions up to 50 (line 302 and Appendix 8), despite not using any gradients. We believe that the advantages we observed for our MCMC method also extend to higher dimensional spaces. In Appendix 9, we describe the results of applying MCMC to a 6400 dimensional groundwater flow model using a diagonal covariance matrix for the proposal distributions. This model is non-differentiable with a highly challenging posterior. Our MCMC method significantly outperforms the other MCMC methods.

As another experiment, we simulate a synthetic linear regression dataset following the procedure in our paper (lines 241-247) in a 1000 dimensional space. Each entry of the feature matrix $\mathbf{X}$ is first sampled i.i.d. from $\mathcal{N}(0,1)$ and then each column $j$ of $\mathbf{X}$ is scaled by the exponential of $2 * \text{rand}(\mathcal{N}(0,1))$. Finally, $\mathbf{y} \sim \mathcal{N}(\mathbf{X}\beta + \beta_0, 100^2)$. We compute the ground truth posterior mean $\mu^* \in \mathbb{R}^{1000}$ by a long run of NUTS, a gradient-based sampler. We run several MCMC methods for a fixed amount of time, and after a burn-in phase, use the samples to estimate the posterior mean, $\widehat{\mu} \in \mathbb{R}^{1000}$. We compare the MCMC methods by the mean-squared error (MSE) between $\widehat{\mu}$ and $\mu^*$ in Table 1. SA-MCMC outperforms MH and AM. SA is more than 10x more sample efficient than AM since SA can achieve a lower MSE with 100,000 likelihood evaluations than AM with 1,000,000 likelihood evaluations. While computing the substitution probabilities in SA-MCMC is time-consuming, the sample efficiency of SA-MCMC can be crucial for problems where the likelihood evaluation is much more expensive, such as simulations in reinforcement learning.

Since SMC is designed for sequence problems and cannot be directly applied, we compared our method with Population Monte Carlo (PMC) [35] which is an iterated importance sampling method with connections to SMC. At each iteration, we sample $N$ particles using the same proposal distribution as in our paper, $q(\cdot) = \mathcal{N}(\cdot|\mu(S), \text{diag}(\Sigma(S)))$. For each particle $x_n$, we compute the weight $w_n = p(x_n)/q(x_n)$ which is used in the estimator. Finally, we resample a set of $N$ unweighted particles by multinomial resampling based on the weights $w_n$. We applied PMC to the logistic regression datasets in our paper. We initialized the $N$ particles from $\mathcal{N}\left(0, \sigma_{q_0}^2 \mathbb{I}\right)$ where we tuned $\sigma_{q_0} \in \left\{10^i, 3 * 10^i\right\}$ for $i \in \{-2,-1,0,1\}$. On 7-dim census, PMC with $N = 100$ and any $\sigma_{q_0}$ led to the $N$ particles becoming identical; sampling from a zero variance Gaussian then raised an error. While PMC with larger $N$ was able to accurately represent the posterior mean in 1 million likelihood evaluations for a few choices of $\sigma_{q_0}$, we found PMC was very unstable. Out of the 5 random runs for each hyperparameter, only 2 of the 5 runs for $(N, \sigma_{q_0}) = (500, 0.3)$, 2 of the 5 runs for $(N, \sigma_{q_0}) = (500, 1.0)$, and 1 of the 5 runs for $(N, \sigma_{q_0}) = (1000, 3.0)$ succeeded out of all the runs. On 11-dim MNIST, we find that PMC failed to estimate the posterior for all choices of $N \in \{100, 500, 1000\}$ and $\sigma_{q_0}$ within 1 million likelihood evaluations. The best log probability of the mean of the particles was -3197 for $(N, \sigma_{q_0}) = (1000, 0.3)$ while the log probability of the typical sample from the posterior is around -2490. Note that all of the MCMC methods in our paper are able to accurately estimate the posterior in less than 100,000 likelihood evaluations. In Appendix 7, we present visualizations of the posterior distributions. Given the narrow and sharply peaked posterior, PMC suffers from a weight degeneracy problem (like particle filters) where almost all of the weight is concentrated on a few particles. This leads to highly inaccurate estimates of $\mu(S), \text{diag}(\Sigma(S))$ and very inefficient proposals. We believe that PMC as importance sampling suffers significantly from the curse of dimensionality: PMC does not work for our 11-dim MNIST.

**Reviewer 2:** SA-MCMC uses a "global" proposal distribution like IMH but unlike many MCMC methods. While SA-MCMC is very good at adapting the proposal distribution, SA-MCMC will not work well when the target distribution cannot be approximated well by *any* member in our family of proposal distributions. Specifically, the assumption that the proposal densities which best approximate the target having uniformly heavier tails than the target is important (lines 179-191). We find SA-MCMC is extremely robust to a poorly chosen initial distribution in the 11-dim MNIST example: for any value of $\sigma_{q_0}$ from $10^{-3}$ to $10^1$, SA-MCMC works perfectly (Figure 5, lines 292-300). In a 1000 dimensional linear regression example, SA-MCMC works with $\sigma_{q_0} = 1$ and 100,000 iterations (see our reply to Reviewer 1).

**Reviewer 3:** The suggestion to have our proposal family be a family of mixture distributions and for our method to learn the optimal mixture distribution (within a given family) is a very important future direction. We will elaborate on the discussion and connection with the Normal Kernel Coupler in our revision. "Parallel Metropolis-Hastings Coupler" is an interesting future direction. We will add the references and discuss future directions in our revision.

[Meta-Review · NeurIPS 2019]

The authors should be commended on writing and submitting a interesting paper on an important topic: adaptive MCMC with an "active" set of samples for estimating a proposal distribution online. The ergodicity theoretical contribution outweighed the deficiencies noted by the reviewers. It would be great if the authors, when working towards the camera-ready, heed the reviewer advice; particularly as it relates to covering related work.